# EXPOSING AND DEFENDING THE ACHILLES' HEEL OF VIDEO MIXTURE-OF-EXPERTS

**Songping Wang**[1*] **Qinglong Liu**[1*] **Yueming Lyu**[1†] **Ning Li**[2] **Ziwen He**[3] **Caifeng Shan**[1†]

[1]Nanjing University [2]China Mobile Information Technology Co., Ltd.
[3]Nanjing University of Information Science and Technology

## ABSTRACT

Mixture-of-Experts (MoE) has demonstrated strong performance in video understanding tasks, yet its adversarial robustness remains underexplored. Existing attack methods often treat MoE as a unified architecture, overlooking the independent and collaborative weaknesses of key components such as routers and expert modules. To fill this gap, we propose **T**emporal **L**ipschitz-**G**uided **A**ttacks (**TLGA**) to thoroughly investigate component-level vulnerabilities in video MoE models. We first design attacks on the router, revealing its independent weaknesses. Building on this, we introduce **J**oint **T**emporal **L**ipschitz-**G**uided **A**ttacks (**J-TLGA**), which collaboratively perturb both routers and experts. This joint attack significantly amplifies adversarial effects and exposes the Achilles' Heel (collaborative weaknesses) of the MoE architecture. Based on these insights, we further propose **J**oint **T**emporal **L**ipschitz **A**dversarial **T**raining (**J-TLAT**). J-TLAT performs joint training to further defend against collaborative weaknesses, enhancing component-wise robustness. Our framework is plug-and-play and reduces inference cost by more than 60% compared with dense models. It consistently enhances adversarial robustness across diverse datasets and architectures, effectively mitigating both the independent and collaborative weaknesses of MoE.

## 1 INTRODUCTION

As the core of artificial intelligence, deep learning has advanced rapidly in recent years (Zhou et al. (2025); Meng et al. (2023); Chen et al. (2025b;c); Han et al. (2026a;b); Liu et al. (2024; 2025)). Central to this progress has been the continuous exploration of scalable and efficient model designs. Mixture-of-Experts (MoE) has emerged as an efficient and powerful deep learning architecture within this field. By introducing a routing mechanism that activates only a small subset of expert sub-networks for each computation, MoE increases model capacity while keeping inference costs under control (Jacobs et al. (1991); Jordan & Jacobs (1994); Shazeer et al. (2017)). This architectural advantage is particularly salient in the domain of video understanding. Video data, characterized by its complex spatial-temporal structures and long-range dependencies, present a fundamental challenge for traditional dense models in balancing expressive power with computational feasibility. MoE addresses this challenge by dynamically selecting experts conditioned on frame-level semantics, thereby demonstrating a remarkable ability to capture motion dynamics and temporal contexts. Consequently, MoE has achieved outstanding performance in tasks such as action recognition and video-language modeling (Jain et al. (2024); Wu et al. (2024); Shaabana et al. (2023)).

While video MoE models have achieved strong performance, their robustness and security have not received commensurate attention. Like other deep learning models, they remain susceptible to adversarial examples—carefully crafted perturbations that can cause high-confidence misclassifications (Goodfellow et al. (2014); Xie et al. (2017); Wang et al. (2025c; 2024a; 2025d;a;b); Miao et al. (2022)). Such vulnerabilities pose serious risks to safety-critical video applications such as autonomous driving and surveillance (Kong et al. (2025); Chen et al. (2022); Wang et al. (2024b)).

Specifically, the vulnerabilities of video MoE models are particularly complex. On one hand, the strong temporal structure of video data allows perturbations to propagate across frames, leading to

---

*Equal contribution. †Corresponding authors. Code is released at https://github.com/DeepSota/J-TLAT.

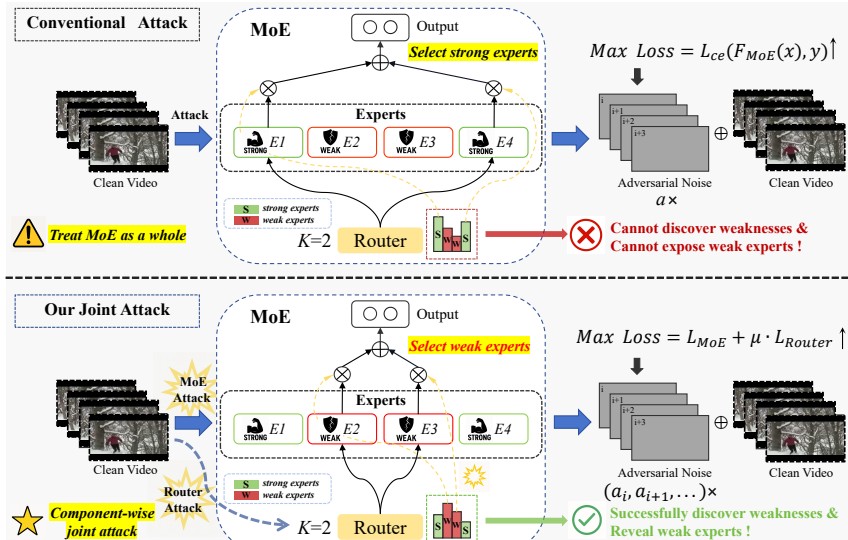

Figure 1: Conventional attacks vs. ours. Conventional attacks treat the MoE as a whole, overlooking its internal architecture and thus capping the attack strength. Our method performs a component-wise joint attack that explicitly targets the Router and Experts, steering the Router toward weak experts and simultaneously perturbing them. This strategy precisely exposes the Achilles' Heel (collaborative weaknesses) of MoE while significantly elevating the attack potency.

cumulative temporal errors. On the other hand, the modular design of MoE—featuring diverse experts and dynamic routing—introduces additional attack surfaces. Adversaries may target individual experts or routers, or disrupt their coordination. However, existing adversarial attacks and training strategies treat MoE as a whole, neglecting the collaborative vulnerabilities of internal components, particularly in the video domain. This highlights the urgent need to discover component-level vulnerabilities in video MoE and to develop specialized adversarial training (AT) mechanisms. We thus ask:

> **(Q.A) What weaknesses do adversarial attacks expose in video MoE models?**
> **(Q.B) Given these insights, how can we develop effective adversarial training to defend against these weaknesses?**

To address the above questions, we first investigate the component-level vulnerabilities of video MoE. We propose a family of Temporal Lipschitz-Guided Attacks (TLGA) that incorporate both Lipschitz regularity and temporal adaptive step-size, targeting the router, the experts, and the MoE model as a whole. We are the first to show that TLGA applied to the router can cause a collapse of routing decisions, posing a serious threat to the model's integrity. Furthermore, we introduce a Joint Temporal Lipschitz-Guided Attack (J-TLGA) targeting both router and experts, showing that coordinated attacks can lead to more destructive outcomes due to inter-component dependencies. J-TLGA reveals that collaborative weaknesses are the Achilles' heel of MoE.

Building on the weaknesses explored by TLGA, we propose two defense strategies: **T**emporal **L**ipschitz **A**dversarial **T**raining (**TLAT**) and **J**oint **T**emporal **L**ipschitz **A**dversarial **T**raining (**J-TLAT**). TLAT strengthens whole MoE robustness by injecting TLGA, while J-TLAT hierarchically enhances adversarial robustness from component-level to overall-level, defending the Achilles' Heel of MoE. Furthermore, we provide a theoretical derivation of the Lipschitz upper bound for MoE in Appendix, which offers theoretical support for our joint attack-defense framework. Our contributions are summarized as follows:

- **component-level vulnerabilities analysis:** We conduct the first systematic investigation into the component-wise vulnerabilities of video MoE, revealing both independent and collaborative failure patterns between the router and experts.
- **Temporal Lipschitz-Guided attacks:** We propose TLGA, which first exposes the vulnerabilities of the routing mechanisms in video MoE. We further extend it to J-TLGA, uncovering the coordinated vulnerabilities between routers and experts.

- **Weakness-guided defense mechanisms:** Based on the identified weaknesses, we introduce two adversarial training frameworks: TLAT enhances global robustness through TLGA, while J-TLAT enhances component-wise and overall robustness hierarchically by addressing their collaborative weaknesses.

- **Extensive evaluation across datasets and models:** Experiments demonstrate J-TLAT effectively defends the Achilles' Heel of MoE, while maintaining inference efficiency and accuracy. We also provide theoretical analysis for the Achilles' Heel of MoE in Appendix.

## 2 RELATED WORK

### 2.1 ADVERSARIAL ATTACK AND DEFENSE ON VIDEO

Recent research shows that video recognition models remain vulnerable to adversarial perturbations. Wei et al. (2019) introduce 3D sparse perturbations to generate adversarial examples in white-box settings, while Wei et al. (2020) employ heuristic-based optimization to perturb selective key frames with minimal noise. Yin et al. (2023) incorporate video transformations into the loss function to enhance attack resilience. Wei et al. (2023) further reduce spatiotemporal redundancy with AstFocus, simultaneously targeting critical frames and regions across inter-frames and intra-frames. Yet, no adversarial attack methods have been specially designed for the promising MoE.

Research on video defense mechanisms remains limited, increasing security risks in real-world applications. OUDefend (Lo et al. (2021)) designs a restoration network against adversarial videos, yet its evaluation is limited to black-box settings. AAT (Kinfu & Vidal (2022)) combines curriculum and adaptive adversarial training to improve robustness against diverse attacks. Yi et al. (2023) boosts robustness via temporal coherence. However, defense for video MoE also remains underexplored.

### 2.2 MIX OF EXPERTS

Mixture-of-Experts (MoE) activates only top-$k$ sub-networks per input (Jacobs et al. (1991); Wang et al. (2020); Fedus et al. (2022)). Despite its widespread use in video applications (Ma et al. (2025)), MoE's resilience to adversarial attacks remains underexplored: Zhang et al. (2023) decomposes MoE robustness into the robustness of the router and experts, yet without designing attacks specifically for MoE, leaving its inherent weaknesses largely unexposed. Similarly, Han et al. (2024) is confined to CNN models, limiting its applicability. Zhang et al. (2025) proposes a dual-model approach for image-based MoE, achieving a more effective trade-off between robustness and accuracy. However, existing methods are tailored to the image domain, leaving attacks on video-based MoE as an open problem. Moreover, the robustness of video MoE is complex due to its unique structure; treating it as a whole while neglecting the interactions between its components makes it difficult to identify the Achilles' Heel. Additionally, videos have more complex features than images, further complicating the development of adversarial attack methods for video MoE. Addressing these robustness issues is crucial for the deployment of MoE in safety-critical applications. This motivates us to conduct systematic research on both attack and defense strategies for MoE.

It should be noted here that the subsequent methods are all configured as white-box attacks. We are committed to designing potent attacks against MoE to maximally expose its fundamental weaknesses and, based on these findings, develop effective defense strategies. As a more powerful attack than gray-box and black-box attacks, a white-box attack (which has knowledge of the target's architecture, parameters, and gradients) can probe its internal vulnerabilities more deeply. Furthermore, if a defense method can enable a model to withstand powerful white-box attacks, it can typically also better defend against weaker gray-box and black-box attacks. This motivates us to design an integrated attack and defense framework for video MoE to ensure its security.

## 3 PRELIMINARIES

**MoE Architecture for Videos.** MoE is a neural network paradigm in which traditional dense layers are replaced by a pool of specialized, sparsely-activated sub-networks—called experts. A lightweight router dynamically selects only a handful of experts for each input, allowing the model's capacity to grow dramatically while keeping the inference cost nearly constant. Video MoE takes

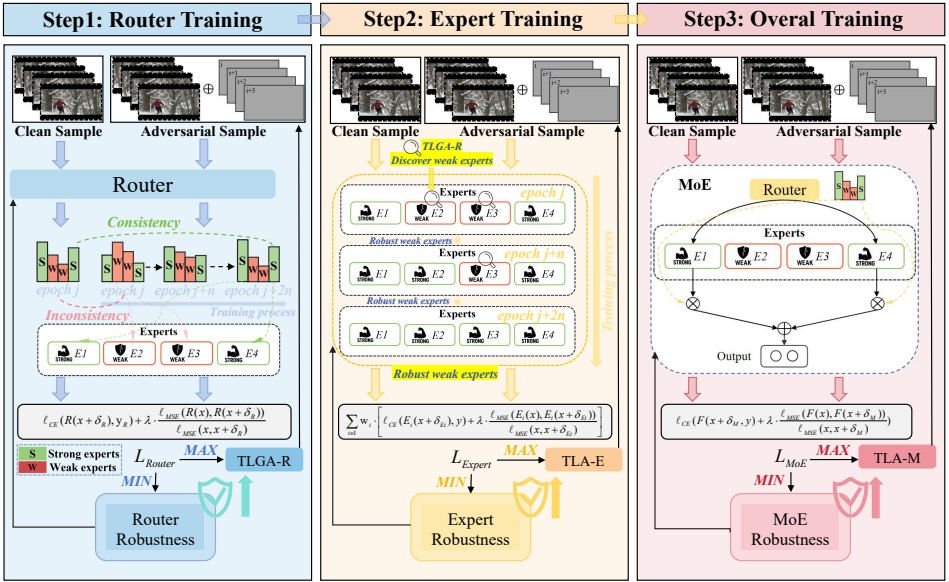

Figure 2: Framework of J-TLAT. **T**emporal **L**ipschitz **G**uided **A**ttack for **R**outer (**TLGA-R**), **T**emporal **L**ipschitz **A**ttack for **E**xpert (**TLA-E**), **T**emporal **L**ipschitz **A**ttack for **M**oe (**TLA-M**) all are designed for MoE. J-TLAT enhances component-wise and overall robustness hierarchically across three steps per epoch.

video data as input, which extends images by a temporal axis. A video clip can be represented as $x \in \mathbb{R}^{C \times T \times H \times W}$, where $C$ is the number of channels (e.g. RGB), $T$ is the number of frames, $H \times W$ is the spatial resolution of each frame. Video MoE consists of Experts $\{E_1(\cdot), \ldots, E_M(\cdot)\}$. Expert $i$ maps the video data to logits $z_i = E_i(x) \in \mathbb{R}^N$. Router $R(\cdot)$ that produces weights for each expert: $R(x) = (w_1(x), ..., w_M(x))$. The final prediction of MoE is the weighted combination of expert outputs: $F(\mathbf{x}) = \sum_{i=1}^{M} w_i(\mathbf{x}) E_i(\mathbf{x}) \in \mathbb{R}^N$. Following several recent works (Videau et al. (2024); He et al. (2024); Chen et al. (2025a)), we use a light network as router $R(\cdot)$ to assign weights to various experts for a given clip. Expert $E_i(x)$ is a neural network to capture specific spatio-temporal content of the video. More details can be seen in Appendix.

**Adversarial Attacks on Video MoE.** Deep networks are inherently brittle, readily succumbing to adversarial perturbations; video-MoE, built upon this foundation, is inevitably exposed to the same threat. An adversarial perturbation $\delta \in \mathbb{R}^{C \times T \times H \times W}$ aimed at attacking video MoE is optimized to maximize the loss under an $\ell_p$-norm budget:

$$\delta^* = \arg\max_{\|\delta\|_p \leq \epsilon} \ell_{CE}\big(\sum_{i=1}^{M} w_i(\mathbf{x} + \delta) E_i(\mathbf{x} + \delta), y\big), \tag{1}$$

where $y$ is the ground-truth label and $\ell_{CE}$ denotes the cross-entropy loss. The inequality $\|\delta\|_p \leq \epsilon$ constrains the $p$-norm of adversarial noise $\delta$ to be less than or equal to a threshold $\epsilon$. We refer to these as conventional attacks, which treat MoE as a whole. Such attacks are not designed for MoE and thus ignore its key components (routers and experts). However, we know that the router and the experts are the core components of MoE. The two work together to realize the core idea of divide and conquer in the MoE architecture. First, treating MoE as a whole and ignoring its core components may limit the effectiveness of the attack. Second, there is currently a lack of attacks designed specifically for the Video MoE architecture. These motivate us to explore attacks targeting the core components of Video MoE.

**Adversarial Attacks on Router and Experts.** To further explore attacks targeting the MoE architecture, focusing on its two core components, we set up two types of component-level attacks, including router attacks and expert attacks. For router attacks, their goal is to alter the routing decisions by attacking the router, thereby increasing the output difference between adversarial samples and clean samples: $\delta_R = \arg\max_\delta \ell(R(x + \delta), R(x))$. For expert attacks, their goal is to produce erroneous outputs by attacking the experts: $\delta_E = \arg\max_\delta \ell_{CE}(E_i(x + \delta), y)$. Our subsequent research demonstrates the effectiveness of component-level attacks: generating adversarial samples

by attacking experts and routers can successfully threaten MoE models and cause them to produce incorrect outputs.

# 4 METHODOLOGY

## 4.1 TEMPORAL LIPSCHITZ-GUIDED ATTACK

The current lack of attacks targeting the Video MoE architecture raises the question:

> **(Q1) How to design attacks against the core components of Video MoE to reduce its robustness and discover its weaknesses?**

Our goal is to undermine the robustness of the model components to perform successful attacks. Lipschitz constant $K$ is a measure of the sensitivity of a function $g$ to changes in its input, effectively reflecting adversarial robustness (Pauli et al. (2021); Zhang et al. (2022); Zühlke & Kudenko (2025)). For a function $g : \mathbb{R}^d \to \mathbb{R}^k$ defined on a domain $\text{dom}(g) \subset \mathbb{R}^d$, there exists a constant $K > 0$ such that for all $x, x + \delta \in \text{dom}(g)$, the following inequality holds:

$$\|g(x) - g(x + \delta)\|_p \leq K \|x - (x + \delta)\|_p, \tag{2}$$

Lipschitz constant reflects the model's sensitivity to minor perturbations. The more robust the model is, the less sensitive it is to minor perturbations, and the smaller the Lipschitz constant. Conversely, the more vulnerable the model is, the more sensitive it is to minor perturbations, and the larger the Lipschitz constant. This motivates us to design attacks by considering the Lipschitz property, in order to increase the Lipschitz constant of MoE, making it more sensitive to minor perturbations and thereby compromising its robustness.

The Lipschitz constant quantifies a model's sensitivity to small perturbations. The more robust a model is, the less sensitive it is to minor perturbations, resulting in a smaller Lipschitz constant. Conversely, a more fragile model exhibits higher sensitivity to such perturbations, leading to a larger Lipschitz constant. This insight motivates us to design an attack by leveraging the Lipschitz property to increase the Lipschitz constant of a Mixture-of-Experts (MoE) model. This makes the model more sensitive to small perturbations, thereby compromising its robustness.

We formulate this idea into a differentiable and optimizable loss objective, denoted as $\mathcal{L}_{\text{Lip}}$. It serves as a direct finite-difference approximation of the local Lipschitz constant and is mathematically expressed as:

$$\mathcal{L}_{\text{Lip}} = \left( \frac{\|g(x + \delta) - g(x)\|_2}{\|\delta\|_2} \right)^2 = \frac{\|g(x) - g(x + \delta)\|_2^2}{\|x - (x + \delta)\|_2^2} = \frac{\ell_{\text{MSE}}(g(x), g(x + \delta))}{\ell_{\text{MSE}}(x, x + \delta)} \tag{3}$$

Here, $g$ can represent the entire model $F$, the router $R$, or an expert $E_i$. By maximizing $\mathcal{L}_{\text{Lip}}$, we can proactively search for directions in the input space that cause the most significant change in the model's output, i.e., directions that maximize the local Lipschitz value. Furthermore, this loss term can be minimized to enhance the model's defensive capabilities. Specifically, minimizing $\mathcal{L}_{\text{Lip}}$ directly penalizes and suppresses the model's steepness in its most vulnerable directions. This process effectively smooths the decision boundary and lowers the upper bound of the global Lipschitz constant, a concept we will discuss further in subsequent sections.

Additionally, videos have an extra temporal dimension. Previous methods set the attack step size for each frame to be the same, ignoring the differences between frames, which limits the effectiveness of the attack. Therefore, we consider utilizing the gradient differences between frames to adaptively allocate attack step sizes for each frame, thereby enhancing the destructiveness from the temporal perspective. Based on the above considerations, we design the Temporal Lipschitz Attack (TLA) to undermine the robustness of the Video MoE architecture from both the temporal and noise sensitivity aspects, as shown in the following formula:

$$\ell_{MoE} = \ell_{CE}(F(x + \delta_M), y) + \lambda \cdot \frac{\ell_{MSE}(F(x), F(x + \delta_M))}{\ell_{MSE}(x, x + \delta_M)}, \tag{4}$$

$$V_{t+1} = \beta \cdot V_t + \|\nabla_x \ell_{MoE}(t)\|_2, \tag{5}$$

$$\alpha^* = \frac{\alpha \cdot \epsilon}{1 + \log(1 + \sqrt{V^*})}, \tag{6}$$

$$x_{adv} = Proj(x + \alpha^* \cdot sign(\nabla_x \ell_M)), \tag{7}$$

where $\ell_{MSE}$ is the MSE loss, $\lambda$ is used to balance the cross-entropy loss with the Lipschitz regularization, $\nabla_x \ell_{MoE}(t)$ represents the gradient of the $t$-th frame, $V^* = (V_1, \ldots, V_T)$ is the temporal momentum adjusted by $\beta$, used to consider the cumulative effect of previous gradient norms, $\alpha$ is the base attack rate, $\alpha^* = (\alpha_1, \ldots, \alpha_T)$ represents the adaptive temporal step size, $Proj(\cdot)$ denotes limiting the perturbation within the $L_{\inf}$-norm. $sign(\cdot)$ denotes sign function and $x_{adv}$ represents the adversarial sample. Eq. 6 is used to smoothly suppress large gradients along the temporal dimension. In order to specifically launch effective attacks against the router, we design the Temporal Lipschitz Router Attack (TLA-R), the formulas can be expressed as:

$$\ell_{Router} = \ell_{CE}(R(x + \delta_R), y_R) + \lambda \cdot \frac{\ell_{MSE}(R(x), R(x + \delta_R))}{\ell_{MSE}(x, x + \delta_R)}, \tag{8}$$

where $y_R$ denotes the index of selected expert for clean sample. By deeply coupling the Lipschitz derivative with the temporal dimension, we enable the attack to not only search for the steepest direction that maximizes the local Lipschitz value, but also to intelligently allocate perturbation resources across the time dimension. This strategy ensures effective achievement of the attack objective.

The remaining process of **T**emporal **L**ipschitz **A**ttack for **R**outer (**TLA-R** ) is similar to Eq. 5-Eq. 7, the same applies to the attacks in the following sections. We use different component-level attack methods and perturbation budget $\epsilon$ to attack a MoE model trained by AT. We use 3D ResNet-18 as the experts with top-1 Router. As shown in Tab.1, the conventional attack PGD-R targeting the Router is difficult to exploit the weaknesses of the Router, and the MoE still maintains a robust accuracy of over 42%. However, our TLA-R attack increases the attack strength by nearly 20%, effectively triggering the routing collapse of the MoE model, thereby causing a sharp drop in robust accuracy. Our method demonstrates that component-level attacks targeting the Router alone can severely threaten MoE models trained by traditional AT. Our attack method TLA-E targeting the expert also outperforms the traditional expert attack PGD-E.

We further speculate that for the same sample, the expert allocated by the Router with higher confidence may be stronger, while the one allocated with lower confidence may be more vulnerable. We can design attacks to guide the Router to assign the sample to the expert with the lowest confidence, thereby exposing the most vulnerable expert and improving attack performance. Based on this, we further design the Temporal Lipschitz Guided Router Attack (TLGA-R), the formula is as follows:

$$\ell^*_{Router} = \ell_{Router} - \gamma_1 \cdot \ell_{CE}(R(x + \delta_R), \hat{y}_R), \tag{9}$$

where $\hat{y}_R$ represents the index of the expert with the lowest confidence output by the Router for the clean sample. As shown in Tab.1, TLGA-R outperforms PGD-R by nearly 24%, achieving the best attack performance for Router attacks. In addition, our method not only demonstrates the feasibility of component-level attacks but also provides a solution for specifically targeting and exposing the weaknesses of the MoE architecture. We have the following insight:

> **Insight 1: By only targeting the Router alone, component-level attacks can severely threaten MoE models trained by traditional AT.**

## 4.2 Joint Temporal Lipschitz-Guided Attack

Based on Insight 1, we further examine the MoE architecture, where the components are not isolated but work together to achieve better performance. Is the same true for attacks? Thus, we raise the following question:

> **(Q2) Can joint attacks be launched against the components of MoE to achieve better attack performance?**

This motivates us to explore the development of joint attacks targeting the MoE architecture. The targets of joint attacks include two aspects: one is the attack on the Router and the other is the attack

on the overall architecture. For the Router attack, we aim to force it to route to the vulnerable experts with low confidence. For the overall attack, we hope to undermine the robustness of the entire MoE architecture and force it to produce incorrect outputs. We adopt the idea of the TLGA-R attack for the Router attack and the idea of the TLA attack for the overall architecture attack. Thus, we design the Joint Temporal Lipschitz Attacks (J-TLA) and Joint Temporal Lipschitz-Guided Attacks (J-TLGA) separately, which can be expressed as followings:

$$\ell_{MoE}^* = \ell_{MoE} + \gamma_2 \cdot \ell_{Router}, \quad \ell_{MoE}^\star = \ell_{MoE} + \gamma_2 \cdot \ell_{Router}^*, \tag{10}$$

where $\ell_{MoE}^*$ and $\ell_{MoE}^\star$ denote the losses for J-TLA and J-TLGA separately. We further conduct experiments with settings consistent with those in Tab.1, comparing our joint attack method with traditional attack methods.

As shown in Tab.1, the joint attack J-TLA significantly reduces the robustness of MoE trained by AT under the budget of $\epsilon = 14/255$ to as low as $4.73\%$. J-TLGA further enhances the attack power. Under J-TLGA, the accuracy of the robust MoE model is only $2.54\%$, which is far superior to the conventional attacks, exposing the Achilles' Heel of MoE. This indicates that: 1) Conventional attack methods are no longer applicable to the MoE architecture. Ignoring its internal components severely limits performance and makes it difficult to deeply reveal the weaknesses of the MoE architecture. 2) The collaboration between components is a double-edged sword, making joint attacks more destructive. Based on these, We have the insight:

> **Insight 2: The vulnerabilities of the components have a cumulative effect. Implementing joint attacks can enhance the attack performance to reveal its weaknesses.**

### 4.3 Joint Temporal Lipschitz Adversarial Training

Traditional adversarial training methods are no longer suitable for the Video MoE architecture. Because of its end-to-end paradigm, it is difficult to finely repair component-level weaknesses and the weaknesses within component collaborations. Its robustness is seriously threatened by joint attacks, and there is an urgent need to develop adversarial training methods more suitable for the MoE architecture based on its weaknesses.

> **(Q3) Can we develop suitable adversarial training mechanisms for Video MoE based on the weaknesses discovered by the attacks?**

The TLA attack undermines the model's robustness by targeting the temporal and perturbation sensitivity dimensions. Conversely, we leverage the adversarial samples generated by TLA to conduct adversarial training on the model, reducing its sensitivity to perturbations and thereby enhancing its adversarial robustness. Based on this, we design Temporal Lipschitz Adversarial Training (TLAT), the formula of which is as follows:

$$\min_{\theta_{MoE}} \max_{x_{adv}} \ell_{MoE}, \tag{11}$$

where $\theta_{MoE}$ denotes the parameters of MoE. Although TLAT can enhance the overall robustness of the Video MoE architecture, it cannot achieve fine-grained robustness improvement at the component level. To realize both fine-grained component-level and overall robustness enhancement of the MoE architecture, we further design Joint Temporal Lipschitz Adversarial Training (J-TLAT). Joint attacks exploit the collaborative weaknesses of components to launch attacks, while J-TLAT repairs the weaknesses discovered by the attacks. As shown in Fig.2, J-TLAT conducts hierarchical robust training in each training epoch. First, it strengthens the robustness of the router to maintain consistent output under different perturbations. Second, based on the TLA-R attack, it identifies the weak experts and conducts adversarial training on them to address the weaknesses. Finally, J-TLAT performs adversarial training on the entire MoE model to further enhance the robustness of the overall collaboration, defending the Achilles' Heel of MoE. The formula is as follows:

$$step1: \min_{\theta_{Router}} \max_{x_{adv}} \ell_{Router}, \tag{12}$$

$$\ell_{Expert} = \sum_{i \in \mathcal{I}} w_i \cdot \left[ \ell_{CE}\left(E_i(x + \delta_{Ei}), y\right) + \lambda \cdot \frac{\ell_{MSE}\left(E_i(x), E_i(x + \delta_{Ei})\right)}{\ell_{MSE}\left(x, x + \delta_{Ei}\right)} \right], \tag{13}$$

$$step2 : \min_{\theta_{Expert}} \max_{x_{adv}} \ell_{Expert}, \ \ step3 : \min_{\theta_{MoE}} \max_{x_{adv}} \ell_{MoE}, \tag{14}$$

where $\mathcal{I} = \text{Top-2}(Router(x_{adv}))$ is the set of weak experts obtained by the TLGA-R attack. $\theta_{Router}$ and $\theta_{Expert}$ denotes the parameters of Router and Experts, respectively. More details for J-TLAT can be seen in Appendix. Based on subsequent experiments, we have the following insight:

> ***Insight 3: Joint adversarial training can enhance overal and local adversarial robustness in a tiered manner by fixing the weaknesses discovered through component-level attacks.***

Table 1: Robust Accuracy (%) under different MoE Attacks with varying perturbation budgets on UCF-101. Robust Accuracy denotes the percentage of correct predictions under attacks.

| $\epsilon$ | Router Attack | | | Expert Attack | | Overall Attack | | | Joint Attack | |
|---|---|---|---|---|---|---|---|---|---|---|
| | **PGD-R** | **TLA-R** | **TLGA-R** | **PGD-E** | **TLA-E** | **FGSM** | **PGD** | **TLA-M** | **J-TLA** | **J-TLGA** |
| 8/255 | 42.20% | 23.52% | **18.13%** | 31.98% | **28.11%** | 30.00% | 22.09% | **15.05%** | 7.03% | **4.95%** |
| 10/255 | 41.98% | 22.31% | **16.48%** | 29.01% | **25.59%** | 27.25% | 17.58% | **12.42%** | 6.81% | **3.41%** |
| 12/255 | 39.78% | 21.98% | **16.22%** | 26.26% | **22.08%** | 25.05% | 15.71% | **11.54%** | 5.39% | **2.75%** |
| 14/255 | 39.34% | 21.65% | **15.82%** | 22.97% | **19.89%** | 24.29% | 14.84% | **11.10%** | 4.73% | **2.54%** |

## 5 EXPERIMENTS

### 5.1 EXPERIMENT SETUP

**Dataset and Recognition Model.** Following Wei et al. (2023), Our experiments are carried out on the widely used UCF-101 (Soomro et al. (2012)) and HMDB-51 (Kuehne et al. (2011)) datasets. Following Wong et al. (2020), we employ three classic models including 3D ResNet-18 (Hara et al. (2018)), TSM (Lin et al. (2019)), Slowfast (Feichtenhofer et al. (2019)) and R(2+1)D (Huang et al. (2021)) as the expert network, utilizing a Top-1 router with four experts for the trade-off between efficiency and accuracy.

**Baselines.** To verify that our methods more effectively exposes the vulnerabilities of MoE, we select classic adversarial robustness benchmarks and extend them to fit video data: FGSM (Goodfellow et al. (2014)), PGD (Madry et al. (2017)) and AutoAttack (Croce & Hein (2020)) as baselines and further incorporate the mainstream white-box video adversarial attack TT (Wei et al. (2022)) as an extended baseline, enabling a more comprehensive comparison. For defensive evaluation, We use the following methods as baselines: AT-S and AT-D (representing sparse and dense expert networks, respectively, both trained by AT), AT-M, OUD-M, and AAT-M (all MoE architectures, trained by AT, OUD (Lo et al. (2021)), and AAT (Kinfu & Vidal (2022)), respectively).

### 5.2 MAIN RESULTS

**Evaluation of the Attack Method.** Experiments in Tab.2 show that TLGA-R attack significantly reduces the robustness of all methods except J-TLAT, highlighting the pronounced weaknesses in the router of the MoE architecture. Under the strong attack of J-TLGA ($\epsilon = 8/255$), the accuracy of OUD-M is 2.64%, and AAT-M even collapses to 0%, demonstrating that joint attacks can effectively reveal the collaborative weaknesses of MoE and greatly enhance the attack effect.

**Evaluation of the Training Method.** Compared with AT-D (dense model), **J-TLAT** achieves a very low Lipschitz constant (0.823). Under all perturbation strengths, J-TLAT shows nearly a 34% improvement over AAT-M under the strongest joint attack J-TLGA, with almost no additional inference cost, strongly demonstrating the effectiveness of the joint training paradigm in enhancing the adversarial robustness of the MoE architecture.

### 5.3 ABLATION STUDY AND MORE RESULTS

**Inner attacks.** Fig.3 compares robust accuracy under FGSM, PGD, and J-TLGA attacks for different adversarial training configurations (PGD, LIP, T-PGD, TLA). LIP denotes the combination

Table 2: Comprehensive evaluation of adversarial robustness on UCF-101 with 3D ResNet as backbone. The table reports clean accuracy (%) and robust accuracies (%) against PGD, AutoAttack, TT, FGSM, TLA-M, TLA-E, TLGA-R, and J-TLGA attacks across four perturbation budgets. Additionally, GFLOPs denotes computational overhead and Lips-R and Lips-J represent the Lipschitz constant values under the TLGA-R and J-TLGA attacks respectively.

| | | Dataset: UCF-101 | Model: 3D ResNet | | | | | |
|---|---|---|---|---|---|---|---|---|
| Method | CLEAN (%) | PGD (%) | | | | AutoAttack (%) | | | |
| | | $\epsilon = 8/255$ | $\epsilon = 10/255$ | $\epsilon = 12/255$ | $\epsilon = 14/255$ | $\epsilon = 8/255$ | $\epsilon = 10/255$ | $\epsilon = 12/255$ | $\epsilon = 14/255$ |
| ○AT-S | 49.89 | 23.85 | 19.45 | 16.81 | 13.19 | 22.75 | 19.45 | 16.26 | 12.97 |
| ○AT-D | 54.51 | 24.84 | 21.98 | 18.57 | 14.73 | 23.30 | 20.11 | 17.25 | 14.18 |
| ○AT-M | 49.23 | 19.23 | 18.46 | 15.93 | 14.62 | 21.43 | 20.00 | 17.69 | 15.82 |
| ○OUD-M | 51.67 | 15.16 | 13.08 | 11.10 | 9.12 | 18.13 | 16.59 | 16.26 | 15.60 |
| ○AAT-M | 49.67 | 23.08 | 20.66 | 19.45 | 18.57 | 22.20 | 19.78 | 18.24 | 13.41 |
| ○TLAT | 51.65 | 30.22 | 27.14 | 22.75 | 19.34 | 22.64 | 20.22 | 19.89 | 18.46 |
| ●J-TLAT | 54.29 | 36.37 | 33.63 | 29.23 | 26.92 | 34.29 | 30.11 | 27.03 | 23.63 |

| Method | GFLOPS ↓ | TT (%) | | | | FGSM (%) | | | |
|---|---|---|---|---|---|---|---|---|---|
| | | $\epsilon = 8/255$ | $\epsilon = 10/255$ | $\epsilon = 12/255$ | $\epsilon = 14/255$ | $\epsilon = 8/255$ | $\epsilon = 10/255$ | $\epsilon = 12/255$ | $\epsilon = 14/255$ |
| ○AT-S | 2.534 | 27.91 | 23.74 | 20.44 | 17.03 | 27.47 | 24.40 | 21.21 | 18.79 |
| ○AT-D | 4.790 | 29.23 | 26.15 | 22.42 | 19.01 | 28.57 | 24.84 | 22.75 | 20.22 |
| ○AT-M | 1.831 | 25.05 | 22.09 | 21.54 | 18.79 | 29.56 | 27.36 | 24.73 | 24.73 |
| ○OUD-M | 19.94 | 22.75 | 20.99 | 20.11 | 17.47 | 24.73 | 24.95 | 22.97 | 23.63 |
| ○AAT-M | 1.831 | 24.51 | 23.41 | 20.44 | 19.34 | 26.92 | 26.59 | 26.48 | 26.37 |
| ○TLAT | 1.831 | 35.16 | 30.99 | 27.03 | 24.84 | 35.16 | 33.08 | 29.56 | 27.69 |
| ●J-TLAT | 1.831 | 37.36 | 34.29 | 30.66 | 28.57 | 38.68 | 36.59 | 33.52 | 32.20 |

| Method | Lips-R ↓ | TLA-M (%) | | | | TLA-E (%) | | | |
|---|---|---|---|---|---|---|---|---|---|
| | | $\epsilon = 8/255$ | $\epsilon = 10/255$ | $\epsilon = 12/255$ | $\epsilon = 14/255$ | $\epsilon = 8/255$ | $\epsilon = 10/255$ | $\epsilon = 12/255$ | $\epsilon = 14/255$ |
| ○AT-M | 261.8 | 18.90 | 16.70 | 15.05 | 15.82 | 29.23 | 27.36 | 22.86 | 19.89 |
| ○OUD-M | 1389 | 8.02 | 6.70 | 7.03 | 5.82 | 22.53 | 19.01 | 16.37 | 13.63 |
| ○AAT-M | 953.3 | 13.41 | 11.43 | 9.45 | 8.24 | 20.55 | 16.70 | 13.19 | 12.09 |
| ○TLAT | 3.500 | 34.51 | 31.21 | 26.04 | 23.30 | 31.98 | 29.12 | 24.95 | 21.10 |
| ●J-TLAT | 0.823 | 34.95 | 31.76 | 28.24 | 24.95 | 34.29 | 31.43 | 25.93 | 23.41 |

| Method | Lips-J ↓ | TLGA-R (%) | | | | J-TLGA (%) | | | |
|---|---|---|---|---|---|---|---|---|---|
| | | $\epsilon = 8/255$ | $\epsilon = 10/255$ | $\epsilon = 12/255$ | $\epsilon = 14/255$ | $\epsilon = 8/255$ | $\epsilon = 10/255$ | $\epsilon = 12/255$ | $\epsilon = 14/255$ |
| ○AT-M | 596.0 | 18.24 | 16.15 | 17.36 | 15.82 | 6.15 | 5.06 | 4.73 | 5.17 |
| ○OUD-M | 1474 | 4.29 | 4.29 | 4.29 | 4.40 | 2.64 | 2.42 | 1.87 | 1.76 |
| ○AAT-M | 1157 | 5.93 | 5.50 | 5.93 | 5.71 | 0.00 | 0.11 | 0.22 | 0.00 |
| ○TLAT | 208.0 | 26.37 | 28.13 | 26.59 | 27.36 | 9.78 | 10.11 | 9.67 | 7.36 |
| ●J-TLAT | 2.343 | 51.87 | 52.42 | 51.76 | 51.87 | 33.96 | 30.33 | 24.95 | 21.98 |

Figure 3: The left figure represents the robustness using different attacks as internal attacks in J-TLAT. The Right figure represents the robustness when using different regularization methods for the external training of J-TLAT on UCF-101.

of PGD attack and Lipschitz regularization, while T-PGD represents the combination of PGD and temporal step-size adaptation. TLA outperforms others, achieving 33.63% robust accuracy under PGD (a 5.5% gain over conventional PGD) and maintaining 30.33% under J-TLGA, highlighting its effectiveness against structured attacks.

**Regularization strategies.** We compare regularization strategies (JS, COS, KL, NR (Tramèr et al. (2017)), GR (Wong et al. (2020))) and **J-TLAT** under three attacks. Here, JS denotes the use of Jensen-Shannon divergence, COS denotes the use of cosine loss, and KL denotes the use of Kullback-Leibler divergence. **J-TLAT** achieves the highest robustness (36.59%, 33.63%, 30.33%) in all scenarios, demonstrating its advantage in enhancing robustness by reducing model sensitivity to perturbations. Fig.4(a) and Fig.4(b) show that our attack methods TLGA-R and J-TLGA achieve better separation of the distributions of adversarial and clean samples against AT-M, thereby realizing superior destructive power. Fig.4(c) and Fig.4(d) illustrate that J-TLAT reduces the IoU distribution difference with higher adversarial robustness. Fig.5 shows that J-TLGA effectively ex-

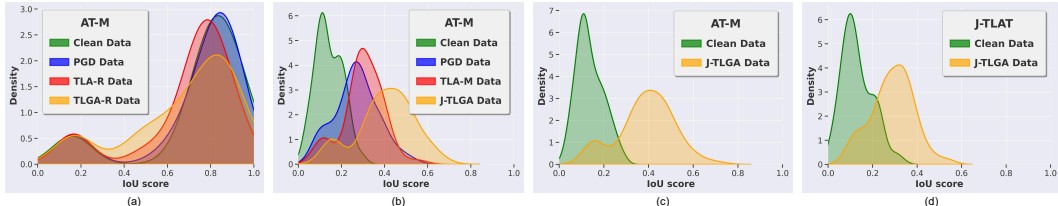

Figure 4: Distribution of IoU scores under different attacks on UCF-101. IoU score is a metric used to measure the output consistency of MoE or Router. The greater the deviation from the IoU distribution of clean data, the higher the attack success rate for the adversarial data.

| Dataset: UCF-101 Model: AT-M Clean Accuracy: 49.78% | | | | |
|---|---|---|---|---|
| Attack | $\epsilon = 8/255$ | $\epsilon = 10/255$ | $\epsilon = 12/255$ | $\epsilon = 14/255$ |
| ○PGD | 22.09% | 17.58% | 15.71% | 14.84% |
| ○LA | 20.66% | 17.24% | 15.38% | 14.73% |
| ○T-PGD | 18.79% | 16.48% | 14.18% | 13.63% |
| ○TLA | 15.05% | 12.42% | 11.54% | 11.10% |
| ●J-TLGA | 4.950% | 3.410% | 2.750% | 2.540% |

Figure 5: Ablation study of our TLGA against AT-M on UCF-101. LA stands for Lipschitz attack, T-PGD stands for PGD combined with temporal adaptive step size.

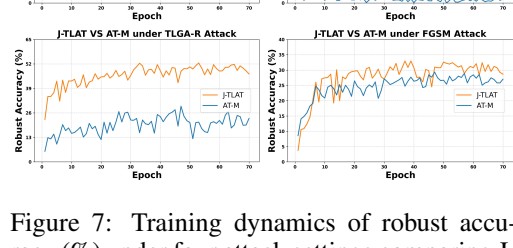

(a) AT-M  (b) J-TLAT

Figure 6: IoU score distributions of AT-M and J-TLAT under TLGA-R attack on UCF-101.

poses the Achilles' Heel of MoE compared with other attacks. Fig.6 illustrates that J-TLAT ensures a more stable routing mechanism under TLGA-R.

| Source Model | Method | 3D Resnet | Slowfast | R(2+1)D |
|---|---|---|---|---|
| 3D Resnet | PGD | 94.80% | 95.00% | 85.61% |
| | FGSM | 69.35% | 95.58% | 87.46% |
| | TT | 66.04% | 94.32% | 84.78% |
| | **J-TLGA** | **20.15%** | **81.79%** | **66.90%** |
| Slowfast | PGD | 87.89% | 77.56% | 88.48% |
| | FGSM | 88.87% | 92.26% | 89.06% |
| | TT | 86.84% | 89.04% | 87.34% |
| | **J-TLGA** | **74.00%** | **49.09%** | **75.91%** |
| R(2+1)D | PGD | 84.64% | 95.57% | 67.90% |
| | FGSM | 87.55% | 96.01% | 84.00% |
| | TT | 84.16% | 94.68% | 78.31% |
| | **J-TLGA** | **66.52%** | **73.59%** | **43.10%** |

Table 3: Transfer attack performance. The values represent the model's remaining accuracy (%) under different transfer attacks. Lower values indicate a more effective attack. J-TLGA possesses better generalization capabilities.

Figure 7: Training dynamics of robust accuracy (%) under four attack settings comparing J-TLAT and AT-M over 70 epochs. J-TLAT consistently achieves higher and more stable robustness across training, maintains strong defense.

**More results.** Fig.3 presents the transfer attack performance of different attacks across various models. Experiments demonstrate J-TLGA outperforms other methods in black-box transfer settings, showing better generalization capabilities. Fig.7 shows AT-M collapses while J-TLAT maintains strong defense across training. More results and details can be found in Appendix.

## 6 CONCLUSION

The MoE architecture has shown great potential in the field of video understanding. However, there is currently a gap in both attacks and defenses methods targeting the video MoE architecture. To address these issues, we propose TLGA to specifically attack the core components of video MoE and further develop the joint attack method J-TLGA, which exposes the Achilles' Heel of video MoE. Based on these weaknesses, we propose the J-TLAT joint training method, which hierarchically enhances the adversarial robustness of MoE architecture from component-level to overall-level. Extensive experiments have demonstrated the effectiveness of our methods, laying the groundwork for future research on adversarial attack and defense in video MoE.

ACKNOWLEDGMENTS

This work was supported by the New Generation Artificial Intelligence-National Science and Technology Major Project (2025ZD0123504), the National Natural Science Foundation of China (Grants 62502200 and 62402228), the Jiangsu Provincial Science and Technology Major Project (Grant BG2024042), and the Natural Science Foundation of Jiangsu Province (Grants BK20251203 and BK20240699).

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
