# OpenReview forum: "Exposing and Defending the Achilles' Heel of Video Mixture-of-Experts"
_ICLR.cc/2026/Conference — ICLR 2026 Poster_

### Official Review · Reviewer_B4KX · 2025-10-30

**Soundness:** 3
**Presentation:** 3
**Contribution:** 3
**Rating:** 4
**Confidence:** 3

**Summary:**

This paper investigates the adversarial robustness of video Mixture-of-Experts (MoE) architectures, an underexplored area despite their growing success in video understanding tasks. The authors propose Temporal Lipschitz-Guided Attacks (TLGA) to analyze vulnerabilities at the router and expert levels, and further introduce Joint TLGA (J-TLGA), which jointly perturbs routers and experts to expose collaborative weaknesses. To mitigate these vulnerabilities, the paper presents Joint Temporal Lipschitz Adversarial Training (J-TLAT), a hierarchical defense framework enhancing both component-wise and overall robustness. Experiments on UCF-101 and HMDB-51 demonstrate that J-TLGA effectively breaks MoE models while J-TLAT substantially improves adversarial robustness with low computational overhead.

**Strengths:**

* Addresses an important and underexplored problem of adversarial robustness in video MoE models.

* Provides novel component-level analysis (router and expert) that exposes previously uncharacterized weaknesses.

* Presents strong empirical results across datasets and models with significant robustness gains and reduced inference cost.

**Weaknesses:**

* Unclear threat model and attacker assumptions

The paper does not explicitly define the threat model used in TLGA and J-TLGA. It is unclear whether the attacks assume a white-box, gray-box, or black-box setting, nor what information the adversary possesses (e.g., access to router parameters, gradients, or architecture details). Since the practicality and interpretability of robustness claims depend heavily on these assumptions, the authors should clearly specify the attacker’s knowledge scope and capabilities. Without this clarification, it is difficult to determine whether the attacks are realistic or merely theoretical.

* Limited diversity of experimental datasets

The evaluation focuses primarily on UCF-101 and HMDB-51, which are relatively small-scale datasets. While these benchmarks are widely used, they may not fully capture the complexity and diversity of modern video understanding tasks. Extending the evaluation to larger and more challenging datasets would significantly strengthen the empirical validity and generalizability of the findings.

* Limited theoretical insight into Lipschitz guidance

The paper briefly introduces temporal Lipschitz coefficients and their role in guiding perturbations, but the theoretical rationale behind why this specific guidance improves attack strength (or defense stability) is not rigorously analyzed. For instance, the connection between the Lipschitz constant and temporal consistency of video features is not deeply discussed. Including theoretical bounds or geometric intuition could clarify how the approach differs from conventional gradient-based or temporal smoothness attacks.

* Defense generalization not fully demonstrated

J-TLAT’s robustness is mainly evaluated against its own attack (J-TLGA). Although it achieves strong performance under this threat, it is unclear whether the defense generalizes to other unseen or non-Lipschitz-based attacks (e.g., FGSM, PGD, or frame-specific perturbations). This makes it difficult to determine whether the method truly enhances general robustness or merely overfits to its own attack pattern. Additional experiments on cross-attack evaluation would greatly improve the paper’s credibility.

**Questions:**

1. What is the attacker’s access level in TLGA and J-TLGA, does the adversary have full gradient access (white-box) or only query access (black-box)?

2. How do hyperparameters (e.g., Lipschitz weighting, perturbation budgets) influence attack effectiveness and defense performance? Are they transferable between datasets?

3. Can J-TLAT improve robustness against non-Lipschitz or unseen attacks? Some ablation in this direction would be informative.

4. Could the component-level attack/defense framework generalize to other MoE domains, such as multimodal fusion or dynamic routing networks?

---

> ### Author Response · Authors · 2025-11-26
>
> ## **Response to Weakness 1 & Question 1: Unclear Threat Model and Attacker Assumptions**
> We appreciate your pointing out this issue and apologize for the lack of clear elaboration in the original manuscript.
>
> Our attack methods TLGA and J-TLGA are both evaluated under a **white-box** setting. We demonstrate the effectiveness of J-TLAT by defending against the most challenging threat model.
>
> Specifically, we assume a powerful attacker possesses the following privileges:
> 1. The attacker understands the entire structure of the MoE model, including the design of the router and the network architectures of all experts.
> 2. The attacker can access and utilize all weights of the model, including the parameters of the router and all expert networks.
> 3. The attacker can fully obtain the gradients of the model.
>
> The purpose of this white-box setting is to evaluate the robustness of our model under the **worst-case scenario** and to deeply explore its internal vulnerabilities. If a defense method enables the model to remain robust against strong white-box attacks, it typically exhibits stronger defensive capabilities against less powerful gray-box and black-box attacks. Our goal is to design powerful attacks targeting MoE to expose its fundamental weaknesses and develop the most effective defense based on these insights. We solemnly commit to revising the revised manuscript to clarify this point in the methodology section.

---

> ### Author Response · Authors · 2025-11-26
>
> ## **Response to Weakness 2: Limited Diversity of Experimental Datasets**
> We appreciate your valuable suggestion regarding dataset selection. We extend our experiments to the larger-scale Jester dataset, which contains nearly 150,000 videos—more than ten times the size of the UCF101 dataset. This allows for a better evaluation of the generalizability of our conclusions.
>
> | Method   | PGD    | FGSM   | TT     | J-TLGA |
> |:---------|:-------|:-------|:-------|:-------|
> | AT-M     | 33.64% | 47.45% | 48.51% | 23.28% |
> | AAT-M    | 40.80% | 59.75% | 62.69% | 26.09% |
> | OUD-M    | 75.89% | 84.54% | 86.42% | 62.43% |
> | J-TLAT   | **80.41%** | **86.23%** | **89.15%** | **71.44%** |
>
> Experimental results on the larger-scale video dataset demonstrate the effectiveness and generalization of our J-TLAT method.

---

> ### Author Response · Authors · 2025-11-26
>
> ## **Response to Weakness 3: Limited Theoretical Insight into Lipschitz Guidance**
> We highly appreciate your profound and constructive comments. Model robustness is closely related to the Lipschitz constant $L$: a small $L$ indicates that the output is insensitive to minor changes in the input. We cleverly translate this insight into a differentiable and optimizable objective.
>
> Our proposed Lipschitz guidance term is not an arbitrary regularizer but a direct finite difference approximation of the local Lipschitz constant, which can be mathematically expressed as:
>
> $$
> L_{Lip} = \frac{L_{MSE}(g(x), g(x+\delta))}{L_{MSE}(x, x+\delta)} = \frac{||g(x) - g(x+\delta)||_2^2}{||\delta||_2^2} = \left( \frac{||g(x+\delta) - g(x)||_2}{||\delta||_2} \right)^2
> $$
>
> Here, $g$ can refer to the entire model $F$, the router $R$, or an expert $E_i$. In the inner maximization process $\max_{\delta}$ of our attack method, we not only search for directions that increase the classification loss $\ell_{CE}$ but also actively seek directions in the input space where the model output changes most drastically (i.e., where the local Lipschitz value is maximized)—a key aspect overlooked by traditional gradient-based and temporal smoothness attacks.
>
> Correspondingly, the outer minimization process $\min_{\theta}$ in the defense component directly penalizes and suppresses the "steepness" of the model in the most vulnerable directions by minimizing the loss incorporating $\mathcal{L}_{\text{Lip}}$, thereby "smoothing" its decision boundary and reducing the upper bound of the global Lipschitz constant.
>
> Regarding the connection between Lipschitz guidance and the temporal consistency of video features, it can be understood as their coupling with the video dimension. Briefly, we treat a video as a spatiotemporal entity $x = (x_1, ..., x_T)$ and deeply couple Lipschitz regularization with the temporal dimension through time-adaptive step sizes.
>
> The core insight is that the gradient norm $\|\nabla_{x_t} \mathcal{L}\|$ of a single frame directly measures its influence on the model’s final prediction: a higher norm indicates that the model is more sensitive to modifications of that frame. This makes frames with high gradient norms the "key frames" of the video sequence—attacking them can cause the maximum prediction deviation with the minimum perturbation budget.
>
> Thus, our "Temporal Lipschitz Guidance" not only searches for the steepest directions that maximize local Lipschitz values (driven by $\mathcal{L}_{\text{Lip}}$) but also intelligently allocates perturbation resources across the temporal dimension, enabling the most efficient achievement of attack objectives. Additionally, we supplement the theoretical derivation of the Lipschitz upper bound in Appendix Section X.

---

> ### Author Response · Authors · 2025-11-26
>
> ## **Response to Weakness 4 & Question 3: Defense Generalization**
> Thank you for your question. We clarify this point: we have conducted extensive cross-attack evaluations in the paper.
>
> In Table 2 of our paper, we assess the robustness of J-TLAT against various standard attacks that are not part of our proposed TLGA/J-TLGA attack series:
> - PGD and FGSM: These are two of the most classic and widely used non-Lipschitz-based gradient attacks.
> - AutoAttack: This is a powerful, parameter-free attack suite regarded as the "gold standard" for evaluating robustness. It includes multiple different types of attacks, effectively avoiding overfitting to a single attack pattern.
> - TT (Temporal Translation): This is a mainstream white-box attack method specifically designed for videos.
>
> The results in Table 2 clearly show that under all these **unseen attacks not proposed by us**, the robustness of J-TLAT is better than other methods. This proves that J-TLAT does not overfit to J-TLGA; instead, it truly fixes the inherent vulnerabilities of the MoE architecture (i.e., collaborative weaknesses), thereby **comprehensively improving its general robustness against various known and unknown attacks**.

---

> ### Author Response · Authors · 2025-11-26
>
> ## **Response to Question 2: Influence of Hyperparameters**
> Thank you for raising the question about the influence of hyperparameters. In fact, detailed ablation studies are included in the appendix.
>
> **1. Influence of hyperparameters:** Appendix C (Figure A.7) and Appendix D (Figure A.8) investigate the impact of attack hyperparameters on attack success rates in TLGA-R and J-TLGA, respectively. Appendix E (Figure A.9) examines the effect of defense hyperparameters in J-TLAT on the model’s standard accuracy (SA) and robust accuracy (RA). Overall, our method exhibits stable performance within a reasonable range of hyperparameters.
>
> **2. Portability:** Our experiments demonstrate that these hyperparameters possess excellent portability. Across all our experiments (spanning UCF-101 and HMDB-51 datasets, as well as 3D-ResNet and TSM backbones, see Tables 2, 4, 5, 6), we adopt a consistent set of hyperparameter configurations and achieve superior performance in all cases. This indicates that they do not require tedious fine-tuning for each specific scenario.

---

> ### Author Response · Authors · 2025-11-26
>
> ## **Response to Question 4: Generalization to Other MoE Domains**
> Thank you for raising this question. In fact, the robustness of MoE can be decomposed into the robustness of the router and the robustness of experts (supplemented with theoretical derivations).
>
> This not only explains the effectiveness of J-TLGA in improving overall attack performance by synergistically undermining the robustness of both routing and expert components (confirming the existence of collaborative weakness) but also validates the rationality of J-TLAT's three-step strategy—only by hierarchically enhancing the component robustness of both the router and expert networks simultaneously, rather than reinforcing a single part in isolation, can system-level robustness be improved from local to global.
>
> This idea is generalizable, as it relates to the inherent divide-and-conquer characteristic of the MoE architecture. It is not limited to the video domain but applies to a wide range of application scenarios:
>
> 1. **Multimodal Fusion**: In multimodal MoE, different experts may process distinct modalities (e.g., images, text). Our method can be used to attack the router and experts, inducing the router to incorrectly select experts with weak processing capabilities or those irrelevant to the target modality, thereby implementing coordinated attacks.
>
> 2. **Dynamic Routing Networks**: These networks also feature a selection mechanism for different experts. Our approach can be leveraged to either attack or enhance the robustness of their core components.
>
> We will explore broader MoE application scenarios in our future work.

---

### Official Review · Reviewer_fCXw · 2025-11-01

**Soundness:** 3
**Presentation:** 3
**Contribution:** 3
**Rating:** 6
**Confidence:** 3

**Summary:**

This paper investigates adversarial robustness of video Mixture-of-Experts (MoE) models. It identifies that conventional attacks treat MoE as a whole and ignore the distinct vulnerabilities of routers and experts. In this paper, the authors propose Temporal Lipschitz-Guided Attack (TLGA)  as a component-level attack on routers or experts and Joint TLGA (J-TLGA) as a coordinated attack exploiting collaborative weaknesses. Based on these two attacks, they finally propose Joint Temporal Lipschitz Adversarial Training (J-TLAT) as a three-stage defense procedure targeting routers, experts, and overall MoE. Experiments on UCF-101 and HMDB-51 show strong improvements in adversarial robustness.

**Strengths:**

- The structure of this paper is clear. The paper is well-written and easy to follow.
- The work analyzes component-level vulnerabilities in video MoE architectures. The idea of differentiating between router and expert robustness is insightful and practically relevant.
- The analysis from attack to defense provides a unified view of robustness analysis. The proposed method provides a comprehensive understanding of the studied problem.
- The experiments are comprehensive, covering multiple backbones (3D-ResNet, TSM, Video Swin) and datasets. The reported reduction in Lipschitz constant and GFLOPs is convincing.
- The appendix provides a clean theoretical derivation for why router manipulation leads to cascading failures.

**Weaknesses:**

- The new attack (TLGA) is basically a standard PGD-style adversarial attack with two small changes: a time-based adjustment of the step size, and an extra term related to the Lipschitz constant. What seems novel is the focus on attacking the router and experts separately in MoE models.
- The loss, e.g., equation (7), is not clear how this actually constrains the Lipschitz constant or enforces smoothness. There’s no clear derivation or estimation of the Lipschitz bound 𝐾.
- The three-stage adversarial training (router → experts → MoE) adds a lot of extra computation. However, the paper only compares inference efficiency, not total training cost. For a fair evaluation, it should compare against other robust training baselines (e.g., AT-M, AAT-M) using the same number of training epochs and attack iterations. Otherwise, the reported robustness gain might simply come from more training rather than the proposed method itself.

**Questions:**

- What the paper calls “collaborative weakness” may actually come from training imbalance. The router tends to favor a few experts while under-training others, so once those experts are attacked, performance drops sharply. This is a common issue in MoE models and can probably be improved by better load balancing.

- The proposed three-stage defense is quite complicated. Some of the same effects, e.g., reducing router over-reliance, might be achieved with simpler methods, such as balanced routing regularization or stochastic expert dropout, without training three times.

- The table layout is incorrect. Some tables are in the middle of the reference..

---

> ### Author Response · Authors · 2025-11-26
>
> ## **Response to Weaknesses 1: Design Philosophy of TLGA**
> Thank you for your affirmation and insightful comments! The idea you recognize—"attacking routers and experts separately in MoE"—is one of the core starting points of our work. The design of TLGA is not a simple minor modification based on PGD, but a collaborative construction around five key ideas of the "video MoE attack-defense loop":
>
> 1. **Temporal-Adaptive Step Size**: Breaks through the limitation of fixed steps in traditional video attacks, dynamically allocates attack intensity according to frame gradient norms, and fully leverages video temporal characteristics to enhance attack effectiveness;
> 2. **Lipschitz Guided Term**: Explicitly maximizes the local Lipschitz constant of the model to accurately amplify the inherent instability of the model and strengthen attack destructiveness;
> 3. **Component-Level Attack Refinement**: Based on TLGA, decouples attack targets into core components such as Router and Expert, successfully revealing component-level vulnerabilities of video MoE (e.g., routing collapse);
> 4. **Joint Attack Upgrade**: Proposes the "collaborative weakness" hypothesis based on component-level weaknesses, and then designs the core attack J-TLGA. By collaboratively attacking routers and experts, it exposes the catastrophic synergistic vulnerabilities of MoE and achieves attack effects far exceeding baselines;
> 5. **Attack-Driven Defense Loop**: Uses TLGA as a vulnerability mining tool and supporting proposes the hierarchical joint defense J-TLAT, comprehensively improving MoE robustness from component to system levels.
>
> These ideas mutually support each other, forming a complete video MoE attack-defense framework of "component-level vulnerabilities detection → joint attack → hierarchical defense".

---

> ### Author Response · Authors · 2025-11-26
>
> ## **Response to Weaknesses 2: Relationship Between the Loss Function and Lipschitz Constant**
> We appreciate your identification of the insufficiency in the explanation of Equation (7). We have provided additional details in Method Section and have supplemented a detailed derivation and estimation of the Lipschitz upper bound in Appendix Section X. Below, we provide a theoretical clarification:
>
> Model robustness is closely related to the Lipschitz constant $L$. A small $L$ indicates that the output is insensitive to minor changes in the input and does not vary sharply with the input, meaning it is smoother. We cleverly translate this idea into a differentiable and optimizable loss objective $\mathcal{L}_{\text{Lip}}$.
>
> It is a direct finite difference approximation of the local Lipschitz constant, which can be mathematically expressed as:
>
>
> $$
> L_{Lip} = \frac{L_{MSE}(g(x), g(x+\delta))}{L_{MSE}(x, x+\delta)} = \frac{||g(x) - g(x+\delta)||_2^2}{||\delta||_2^2} = \left( \frac{||g(x+\delta) - g(x)||_2}{||\delta||_2} \right)^2
> $$
>
> Here, $g$ can refer to the entire model $F$, the router $R$, or an expert $E_i$. In the inner maximization process $\max_{\delta}$ of the attack in our method, we not only search for directions that increase the classification loss $\mathcal{L}_{\text{CE}}$ but also actively seek directions in the input space where the model output changes most drastically (i.e., where the local Lipschitz value is maximized).
>
> Additionally, the outer minimization process $\min_{\theta}$ in the defense component directly penalizes and suppresses the "steepness" of the model in the most vulnerable directions by minimizing the loss that includes $\mathcal{L}_{\text{Lip}}$, thereby "smoothing" its decision boundary and reducing the upper bound of the global Lipschitz constant.

---

> > ### Author Response · Authors · 2025-11-27
> >
> > ## **Response to Weaknesses 2: Derivation and Estimation of Lipschitz Upper Bound**
> >
> > To further address your concerns and consolidate the theoretical foundation, we supplement a derivation of the Lipschitz constant upper bound specifically for Mixture of Experts (MoE) models. This derivation reveals the effectiveness of component-wise joint attacks and defenses.
> >
> > Let $L_R$ denote the Lipschitz constant of the router $R(x)$ , and $L_{E_i}$ denote the Lipschitz constant of each expert network $E_i(x)$. For the detailed derivation, please see our Appendix, Section X. This gives an upper bound on the Lipschitz constant of the entire MoE model:
> > $$
> > L_F \le B_E \sqrt{M} L_R + \max_j L_{E_j}
> > $$
> >
> > This derivation offers solid theoretical support for J-TLAT: the overall robustness of the model ($L_F$) is directly coupled with the smoothness of the router ($L_R$) and the smoothness of the weakest expert network ($\max_j L_{E_j}$).
> >
> > This not only explains the effectiveness of J-TLGA in undermining system robustness by collaboratively increasing $L_R$ and $\max_j L_{E_j}$ (validating the existence of collaborative weakness) but also justifies the rationality of J-TLAT’s three-step strategy—component-wise robustness of both the router and expert networks must be improved hierarchically and simultaneously, rather than reinforcing a single component in isolation. Only through this approach can $L_F$ be reduced from local to global levels, achieving system-level robustness enhancement.
> >
> > Regarding the empirical estimation of $K$: In fact, the Lips-R and Lips-J values reported in Table 2 of the main text are posterior empirical estimates of local Lipschitz constants under our specific attack settings. The significant decrease in these empirical values achieved by our J-TLAT method indicates that our training process successfully enforces smoothness and constrains the actual Lipschitz constant.

---

> ### Author Response · Authors · 2025-11-26
>
> ## **Response to Weakness 3: Computational Cost and Fairness**
> We fully understand your concerns. For a fairer comparison, we uniformly set the number of internal attack iterations for different methods to 6, conduct comparative experiments on UCF101 and 3D ResNet, and perform a more comprehensive evaluation of adversarial robustness, training GFLOPS, total training time, and other metrics.
>
> It is important to note that our J-TLAT executes three steps within each epoch. Each step involves two iterative attacks on three components, totaling six attacks (Router → Expert → MoE), which is more efficient than three isolated stages.
>
> | Method | PGD | FGSM     | TT       | J-TLGA   | GFLOPS  | Time (Min) |
> | ------ | ------- | -------- | -------- | -------- | ------- | ---------- |
> | AT-M   | 19.79%  | 29.08%   | 25.22%   | 6.27%    | 1.831   | 130.6      |
> | AAT-M  | 23.30%  | 27.22%   | 25.16%   | 0.33%    | 1.831   | 142.2      |
> | OUD-M  | 18.82%  | 25.15%   | 23.85%   | 2.89%    | 19.94   | 226.3      |
> | J-TLAT | **36.37%** | **38.68%** | **37.36%** | **33.96%** | **1.831** | **108.5**  |
>
> Experiments show that the advantage of J-TLAT does not simply come from "more training". J-TLAT differs fundamentally from traditional adversarial training strategies. Traditional adversarial training applies a "one-size-fits-all" robustness signal to the entire MoE model, but this signal cannot effectively transmit to the router and the weak experts "hidden" by the router. In contrast, J-TLAT performs hierarchical repair targeting the component weaknesses revealed by TLGA, thereby ensuring robustness from local components to the whole.
>
> Additionally, Appendix Figure 13 shows that J-TLAT not only achieves higher robustness in the end but also quickly surpasses AT-M in the early stages of training. This strongly proves that the effectiveness of J-TLAT stems from its strategy of hierarchical vulnerability repair, rather than mere accumulation of computational load.

---

> ### Author Response · Authors · 2025-11-26
>
> ## **Response to Question 1: Existence of Collaborative Weakness**
> Your insight is profound. We further clarify that the "collaborative weakness" of MoEs is not inherently derived from training imbalance, but rather an intrinsic byproduct of their core "divide-and-conquer" strategy.
>
> Specifically, different experts specialize in distinct domains, and the router assigns data to the most proficient expert based on data characteristics. When J-TLGA diverts data to the least proficient expert (relatively weak expert) for targeted attacks, collaborative weakness are triggered. We introduce a balance loss [1] to conduct adversarial training on MoE models and evaluate their adversarial robustness.
>
> | Eps | PGD | FGSM | TT | J-TLGA |
> |---|---|---|---|---|
> | 0/255 | 52.97 | 52.97 | 52.97 | **52.97** |
> | 8/255 | 18.13% | 25.27% | 26.15% | **8.51%** |
> | 10/255 | 15.05% | 23.41% | 22.53% | **7.41%** |
> | 12/255 | 12.64% | 21.21% | 19.78% | **5.97%** |
> | 14/255 | 10.33% | 19.45% | 16.48% | **5.26%** |
>
> Experimental results demonstrate that while load balance loss alleviates collaborative weakness to a certain extent, its effect is extremely limited.
>
> The problem stems from two aspects: First, experts have distinct specializations (the essence of MoEs). This means every sample has experts that are "proficient" or "unproficient" in handling it; targeted attacks like J-TLGA can easily break the balance, redirect samples to "weak experts," and launch attacks to produce synergistic disruption effects. Second, achieving perfect load balance during model training is challenging, which further exacerbates the relative strengths and weaknesses among experts.
>
> In summary, the root cause of collaborative weakness lies in the fact that as long as specialization differences exist between experts (otherwise, MoEs degenerate into redundant dense models), the possibility of creating "mismatches between experts and tasks" through coordinated attacks persists indefinitely. This constitutes the most fundamental and exploitable attack surface.
>
> [1] Shazeer, Noam, et al. "Outrageously large neural networks: The sparsely-gated mixture-of-experts layer." arXiv preprint arXiv:1701.06538 (2017).

---

> ### Author Response · Authors · 2025-11-26
>
> ## **Response to Question 2: Replacement with Different Load Balancing Strategies**
> Thank you for proposing these insightful alternative schemes.
>
> | Strategy               | PGD    | FGSM    | TT      | J-TLGA  |
> | :--------------------- | :----- | :------ | :------ | :------ |
> | Balancing Loss         | 15.05% | 23.41%  | 22.53%  | 7.41%   |
> | Stochastic Expert Dropping | 21.32% | 25.93%  | 25.93%  | 16.70%  |
> | Routing Regularization | 18.76% | 24.73%  | 23.63%  | 8.18%   |
> | J-TLAT                 | **32.75%** | **35.93%** | **33.85%** | **29.78%** |
>
> As shown in the table, although Stochastic Expert Dropping[1] and Routing Regularization[2] can alleviate the co-vulnerability to a certain extent, they fail to fundamentally address the inherent component-level weaknesses of MoE models.
>
> Our method effectively prevents the "attack chain" of coordinated attacks through three key steps:
>
> **(1) Step 1 (Robustify Router):** Breaks the first link of the attack chain.
>
> **(2) Step 2 (Reinforcing Weak Experts):** Repairs the weakest link in the attack chain.
>
> **(3) Step 3 (Overall Training):** Ensures the entire system can work synergistically after repairs to defend against joint attacks.
>
> Experimental results demonstrate that J-TLAT outperforms all baselines by a significant margin in robustness when facing the strongest J-TLGA attack, confirming the value of this targeted design.
>
> [1] Zuo, Simiao, et al. "Taming sparsely activated transformer with stochastic experts." arXiv preprint arXiv:2110.04260 (2021).
>
> [2] Zoph, Barret, et al. "St-moe: Designing stable and transferable sparse expert models." arXiv preprint arXiv:2202.08906 (2022).
>
> ## **Response to Question 3: Table Layout**
> Thank you very much for carefully pointing out this formatting error! We sincerely apologize for any inconvenience caused. We have revised the manuscript to ensure that it adheres to all formatting guidelines.

---

### Official Review · Reviewer_6k8Y · 2025-11-03

**Soundness:** 2
**Presentation:** 2
**Contribution:** 3
**Rating:** 4
**Confidence:** 4

**Summary:**

This paper is the first to explore adversarial attacks in video mixture-of-experts. Through experiments, the authors highlight component-level vulnerabilities in the video moe structure. Corresponding attacks are designed for both the moe and the router, and corresponding adversarial training is proposed to improve the model's adversarial robustness. Experiments validate the effectiveness of the proposed methods.

**Strengths:**

1. Exploring the adversarial robustness of the video moe structure is beneficial for the secure deployment of the model.

2. This paper proposes an adversarial training method to improve adversarial robustness.

**Weaknesses:**

1. My biggest concern is the generalization ability of the experimental results. The number of models is limited, and the results of multiple models cannot be represented in the main body. Most importantly, the clean accuracy in Table 2 is extremely low, lower than the performance of common models on UCF-101, meaning the models were not well trained. Attacking a model that is prone to misclassification is easier and will lead to problematic conclusions.

2. This paper is unclear in its expression. It involves several self-proposed attacks and defenses, with abbreviations that are merely letter abbreviations without any semantic information, making them easily confusing. Even more egregiously, the abbreviation of Temporal Lipschitz Guided Router Attack as TLGA-R is unclear and doesn't seem to follow any specific principle.

3. The performance representation in Table 2 is misleading. Performance under different attacks should also be compared. Based on the experimental results, TLA-M and TLA-E are inferior to PGD, which weakens the contribution of this work.

4. Current research on adversarial examples focuses more on black-box attacks. This paper involves three models; why are there no experimental results for black-box transfer attacks?

**Questions:**

Please refer to Weaknesses.

---

> ### Author Response · Authors · 2025-11-26
>
> ## **Response to Weakness 1: Generalization Ability and Low Clean Accuracy**
> Thank you for raising this concern. To address your worry, we conduct experiments using additional models and a larger-scale dataset to comprehensively evaluate generalization ability, ensuring all models underwent sufficient adversarial training (with clean accuracy exceeding 90%).
>
> Specifically, we adopt the Jester[1] dataset, which contains more than ten times the data volume of the UCF101 dataset, enabling more robust assessment of generalization performance.
>
> | Eps    | PGD    | FGSM    | TT      | J-TLGA   |
> | :----- | :----- | :------ | :------ | :------- |
> | 0/255  | 94.8%  | 94.8%   | 94.8%   | 94.8%    |
> | 8/255  | 77.75% | 83.81%  | 85.87%  | **69.25%** |
> | 10/255 | 67.28% | 80.37%  | 82.54%  | **51.31%** |
> | 12/255 | 53.14% | 75.49%  | 75.72%  | **33.39%** |
> | 14/255 | 38.41% | 69.23%  | 65.54%  | **19.91%** |
>
> Experimental results demonstrate that our J-TLGA method exhibits advantages across all perturbation levels. As the perturbation budget increases, the attack effectiveness of J-TLGA becomes increasingly prominent, and the performance gap with other methods continues to widen. For instance, under a perturbation of 14/255, J-TLGA reduces the model accuracy to only 19.91%.
>
> This strongly indicates that even for high-precision MoE models with stronger defense capabilities, J-TLGA can more efficiently identify and exploit the deep-seated vulnerabilities of the models. In subsequent responses, we will present experimental results of transfer attacks covering additional models and display them in the main text (Table 3).
>
> [1] Materzynska, Joanna, et al. "The jester dataset: A large-scale video dataset of human gestures." *Proceedings of the IEEE/CVF International Conference on Computer Vision Workshops*. 2019.

---

> ### Author Response · Authors · 2025-11-26
>
> ## **Response to Weakness 2: Unclear Expression and Confusing Abbreviations**
> Thank you for your valuable feedback on the paper's expression. To address your suggestions and fully resolve your concerns, we will provide detailed clarifications and formulate a revision plan. Our naming convention is mainly centered around the core technology: **T**emporal **L**ipschitz **A**ttack (**TLA**), and subdivided according to attack/defense objectives:
>
> 1. Attack Method Series
> - **TLA-M** (**T**emporal **L**ipschitz **A**ttack for **M**odel): Represents the Temporal Lipschitz Attack targeting the overall MoE model.
> - **TLA-E** (**T**emporal **L**ipschitz **A**ttack for **E**xpert): Represents the Temporal Lipschitz Attack targeting Expert.
> - **TLA-R** (**T**emporal **L**ipschitz **A**ttack for **R**outer): Represents the Temporal Lipschitz Attack targeting Router.
> - **TLGA-R** (**T**emporal **L**ipschitz **G**uided **A**ttack for **R**outer): This naming emphasizes the core mechanism of "guiding the router to vulnerable experts" added on the basis of TLA-R.
> - **J-TLGA** (**J**oint **T**emporal **L**ipschitz **G**uided **A**ttack): Indicates that this method is a joint attack targeting multiple key components of MoE.
>
> 2. Defense Method Series
> - **TLAT** (**T**emporal **L**ipschitz **A**dversarial **T**raining)
> - **J-TLAT** (**J**oint **T**emporal **L**ipschitz **A**dversarial **T**raining): Similarly, it represents the joint adversarial training for key components introduced on the basis of TLAT.
>
> Based on the above clear logic, we make detailed revisions in the main text and include a clear "Naming Rules and Semantic Abbreviation Description" in the appendix Section.V for readers' convenient reference. We hope these detailed and comprehensive modifications will completely eliminate ambiguities in naming and enhance the clarity and readability of the paper.

---

> ### Author Response · Authors · 2025-11-26
>
> ## **Response to Weakness 3: Misleading Performance Representation**
> We appreciate your identification of this potential misunderstanding and would like to provide clarifications as follows:
>
> Attack types (Table 1 of the paper) can be categorized into four classes: Router Attack (PGD-R VS. TLGA-R), Expert Attack (PGD-E VS. TLA-E), Overall Attack (PGD VS. TLA-M), and Joint Attack.
>
> In fact, comparisons within the same attack type are more equitable. As shown in Table 1, TLA-E outperforms PGD-E in "Expert Attack," TLGA-R achieves an average improvement of 24% over PGD-R in "Router Attack," and TLA-M surpasses PGD in "Overall Attack"—these results validate the targeted effectiveness of component-level attacks.
>
> Furthermore, the core objectives of designing TLA-M, TLA-E, and TLGA-R are not to pursue the strongest performance in individual scenarios, but to develop them as component-level analytical tools and building blocks. Their key values are reflected in three aspects:
> 1. **Precise Isolation of Component Vulnerabilities**: TLA-E and TLGA-R can independently detect vulnerabilities in expert networks and routers (e.g., TLGA-R can trigger router collapse), laying the foundation for subsequent research;
> 2. **Support for Core Joint Attack J-TLGA**: Based on insights from the aforementioned base attacks, we propose our core contribution—the Joint Attack J-TLGA. It synergistically targets both routers and expert networks to expose their collaborative vulnerabilities, achieving performance far exceeding PGD and all baselines (Tables 1 and 2);
> 3. **Attack-Driven Defense**: We implement hierarchical defense through "Component-Level Joint Adversarial Training (J-TLAT)," ultimately constructing an integrated MoE attack-defense framework encompassing "Component-Level Tools → Joint Attack → Joint Defense."
>
> TLA-M and TLA-E serve as the foundation of this framework rather than the core contributions. The primary value of this paper lies in being the first to propose an MoE collaborative vulnerability attack and hierarchical defense scheme, forming a systematic attack-defense closed loop that effectively enhances the robustness of MoE architectures.

---

> ### Author Response · Authors · 2025-11-26
>
> ## **Response to Weakness 4: Lack of Black-Box Attacks**
>
> This is an excellent observation. To address this, we incorporate additional models as experts in the Mixture of Experts (MoE) framework on the Jester dataset, conduct standard adversarial training, and evaluate the effectiveness of different methods through black-box transfer attack experiments.
>
> | Source Model | Method       | 3D Resnet | Slowfast | R2+1dD  |
> | :----------- | :----------- | :-------- | :------- | :------ |
> | 3D Resnet    | PGD          | 94.80%    | 95.00%   | 85.61%  |
> | 3D Resnet    | FGSM         | 69.35%    | 95.58%   | 87.46%  |
> | 3D Resnet    | TT           | 66.04%    | 94.32%   | 84.78%  |
> | 3D Resnet    | **J-TLGA**   | **20.15%**| **81.79%**| **66.90%**|
> | Slowfast     | PGD          | 87.89%    | 77.56%   | 88.48%  |
> | Slowfast     | FGSM         | 88.87%    | 92.26%   | 89.06%  |
> | Slowfast     | TT           | 86.84%    | 89.04%   | 87.34%  |
> | Slowfast     | **J-TLGA**   | **74.00%**| **49.09%**| **75.91%**|
> | R2+1D        | PGD          | 84.64%    | 95.57%   | 67.90%  |
> | R2+1D        | FGSM         | 87.55%    | 96.01%   | 84.00%  |
> | R2+1D        | TT           | 84.16%    | 94.68%   | 78.31%  |
> | R2+1D        | **J-TLGA**   | **66.52%**| **73.59%**| **43.10%**|
>
> The table presents the model accuracy after being attacked. Experiments demonstrate that even against more robust defense models, J-TLGA outperforms other methods in black-box transfer attacks.
>
> This indicates that J-TLGA possesses stronger generalization capabilities, enabling it to more effectively attack unknown black-box models. It validates the effectiveness of enhancing attack performance by targeting the collaborative vulnerabilities of MoE models.

---

### Public Comment · ~Xu_Zhang43 · 2025-11-17
**Missing Related Work on MoE Robustness**

This paper studies an important topic, namely improving the adversarial robustness of Mixture-of-Experts (MoE) models, which is essential for deploying MoE architectures in reliable and safety-critical applications.

Our recent ICML 2025 work *“Optimizing Robustness and Accuracy in Mixture of Experts: A Dual-Model Approach”* also investigates this problem from a complementary angle. Both studies analyze the vulnerabilities of routers and experts, identify their distinct robustness behaviors, and propose joint or dual training frameworks to enhance overall robustness while maintaining accuracy.

Given the strong alignment between the two studies, it would be valuable for this paper to discuss or compare with that work to clarify the similarities and differences in methodology and findings.

---

> ### Author Response · Authors · 2025-11-26
>
> ## **Response to the Relevant and Insightful ICML 2025 Work**
>
> Thank you for insightful ICML 2025 work. We appreciate your positive assessment of our paper's direction. We agree that there is a strong alignment between our studies, as both investigate the vulnerabilities of MoE models.
>
> To clarify the relationship between our works, we will cite your paper and add a detailed discussion of your paper in our revised Related Work section. We highlight the complementary nature of our approaches: while your work proposes a dual-model approach for image-based MoE, our paper focuses specifically on the video domain, introducing temporal dynamics into attack frameworks (e.g., temporal adaptive step-size). Furthermore, we have conducted component - level theoretical analysis, develop joint attacks to expose the Collaborative Weakness of Mixture of Experts (MoE) models. Building on these attacks, we have promoted the development of defense strategies and proposed a hierarchical joint method that enhances robustness from local components to the entire system.
>
> We believe this discussion will provide readers with a more comprehensive view of the current research landscape on MoE robustness. Thank you again for your constructive feedback. Your work has given us great inspiration.

---

### Author Response · Authors · 2025-11-29

## **General Response by Authors**

We sincerely thank all reviewers 6k8Y, fCXw, and B4KX for investing their valuable time in reviewing our submission. We appreciate your comments, constructive suggestions, and detailed questions, all of which have directly facilitated the clarification and refinement of our manuscript. We genuinely view this process as a collaborative effort and are grateful for the reviewers' engagement with both the high-level motivations and technical details of our proposed methods.

In the reviews, we are pleased to receive several consistent points of positive feedback:
1. **Novelty and Insightfulness**: Reviewers B4KX and fCXw both emphasized this aspect. Specifically, Reviewer B4KX commended the work for being able to "Provide novel component-level analysis," while Reviewer fCXw praised it as "is insightful and practically relevant."
2. **Clarity of Expression and Rigor of Theory**: Reviewer fCXw praised the paper for its clear structure and ease of understanding, and pointed out that the analysis in the paper is comprehensive and the theoretical derivation is logically rigorous.
3. **Practical Value and Contribution**: Reviewer 6k8Y noted that the work "is beneficial for the secure deployment"; Reviewer B4KX, on the other hand, emphasized that it "Addresses an important and underexplored problem."
4. **Comprehensive Experimental Performance**: All reviewers recognized this merit. Reviewer fCXw appreciated the comprehensiveness of the experimental design in this paper, noting that it is convincing; Reviewer B4KX praised the work for being able to "Present strong empirical results across datasets and models."

The constructive suggestions from the reviewers have greatly improved the quality of the paper. In response to their feedback, we have taken the following actions:
To fully address the concerns raised by the reviewers and further enhance the quality of the paper, we have implemented a series of targeted improvements. The key revisions are summarized as follows:
1. Extended the experiments to the larger-scale Jester dataset (10 times the size of UCF101) and included more models. We also ensured the high accuracy of the models to demonstrate their generalization ability and the transferability of black-box attacks (addressing 6k8Y and B4KX).
2. Fully clarified the naming rules of the methods and the performance of the methods, and supplemented explanations of naming rules and semantic abbreviations in the appendix to eliminate misunderstandings (addressing 6k8Y).
3. Strengthened the explanation of the theoretical basis and method effectiveness in the main text, and added a more detailed derivation process in Appendix X (addressing fCXw and B4KX).
4. Clarified the essence of the existence of "Collaborative Weakness" from both theoretical and experimental perspectives (addressing  fCXw).
5. Clarified the defense mechanism of J-TLAT and conducted a variety of comparative experiments to demonstrate its advantages in terms of effectiveness and efficiency (addressing fCXw).
6. Made detailed revisions in the paper to clearly define the threat model and clarify its rationality (addressing B4KX).
7. Clarified the generalization of the defense method and supplemented hyperparameter experiments (addressing B4KX). (In fact, the main paper already includes the unseen attacks mentioned by reviewer, such as PGD and FGSM attacks, and the appendix also contains complete hyperparameter experiments.)

We believe these changes have strengthened the paper, and we appreciate the reviewers’ guidance in arriving at a clearer, more rigorous, and more thoroughly justified presentation.

Once again, we thank all reviewers for their encouragement and help in improving the manuscript.

---

### Meta-Review · Area_Chair_A2jm · 2026-01-10

**Summary:**

This paper investigates the adversarial robustness of Mixture-of-Experts (MoE) models for video understanding. Reviewers find the work novel, insightful, and valuable to the community. Several concerns were raised about the limited diversity of evaluation datasets, the absence of black-box transfer attacks, and low clean accuracy on UCF-101. While the clean-accuracy concern remains, most of these issues were adequately addressed during rebuttal. Overall, the strengths of the work outweigh its weaknesses.

**Reviewer Concerns:**

Reviewer 6k8Y

W1: Generalization ability and low clean accuracy. [still outstanding]

R1: Although the authors show promising results on an additional dataset during rebuttal, the low clean accuracy issue on UCF-101 remains unsolved.

W2: TLA-M and TLA-E are inferior to PGD. [addressed by the rebuttal]

R2: The authors clarify that the core contribution is J-TLGA.

W3: No experimental results for black-box transfer attacks. [addressed by the rebuttal]

R3: The authors conducted experiments on black-box transfer attacks during rebuttal and showed that J-TLGA possesses stronger generalization capabilities.

---
Reviewer fCXw

W1: The novelty is just the focus on attacking the router and experts separately in MoE models. [addressed by the rebuttal]

R1: The authors claim that there are five key innovations of the proposed method.

W2: Confusion about Equation (7). [addressed by the rebuttal]

R2: The authors clarify this point during rebuttal.

W3: No training cost comparison. [addressed by the rebuttal]

R3: The authors conduct additional experiments during rebuttal and show that the proposed method is training efficiently.

---
Reviewer B4KX

W1: Unclear threat model and attacker assumptions. [addressed by the rebuttal]

R1: The authors clarify this point during rebuttal.

W2: Limited diversity of experimental datasets. [addressed by the rebuttal]

R2: The authors conducted experiments on an additional dataset and showed promising results.

W3: Limited theoretical insight into Lipschitz guidance. [addressed by the rebuttal]

R3: The authors provided more analysis during rebuttal.

W4: Defense generalization not fully demonstrated. [addressed by the rebuttal]

R4: The authors clarify that the results are already in Table 2.

**Reviewer Scores:**

Reviewer 6k8Y: 4 -> 4

Reviewer fCXw: 6 -> 6

Reviewer B4KX: 4 -> 6

Average score: 5.33

---

### Decision · Program_Chairs · 2026-01-26

Accept (Poster)